# Dimer description of the SU(4) antiferromagnet on the triangular lattice

**Anna Keselman[1*,2], Lucile Savary[3,1] and Leon Balents[1,4]**

**1** Kavli Institute for Theoretical Physics, University of California,
Santa Barbara, CA 93106-4030
**2** Station Q, Microsoft Corporation, Santa Barbara, California 93106-6105, USA
**3** Université de Lyon, École Normale Supérieure de Lyon, Université Claude Bernard Lyon I,
CNRS, Laboratoire de physique, 46, allée d'Italie, 69007 Lyon
**4** Canadian Institute for Advanced Research, Toronto, Ontario, Canada

★ akeselman@kitp.ucsb.edu

## Abstract

In systems with many local degrees of freedom, high-symmetry points in the phase diagram can provide an important starting point for the investigation of their properties throughout the phase diagram. In systems with both spin and orbital (or valley) degrees of freedom such a starting point gives rise to SU(4)-symmetric models. Here we consider SU(4)-symmetric "spin" models, corresponding to Mott phases at half-filling, i.e. the six-dimensional representation of SU(4). This may be relevant to twisted multilayer graphene. In particular, we study the SU(4) antiferromagnetic "Heisenberg" model on the triangular lattice, both in the classical limit and in the quantum regime. Carrying out a numerical study using the density matrix renormalization group (DMRG), we argue that the ground state is non-magnetic. We then derive a dimer expansion of the SU(4) spin model. An exact diagonalization (ED) study of the effective dimer model suggests that the ground state breaks translation invariance, forming a valence bond solid (VBS) with a 12-site unit cell. Finally, we consider the effect of SU(4)-symmetry breaking interactions due to Hund's coupling, and argue for a possible phase transition between a VBS and a magnetically ordered state.

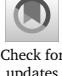

# 1 Introduction

Frustrated quantum antiferromagnets may possess non-magnetic ground states, avoiding spin order through short or long range entanglement of spins. In the late 1980s and early 1990s, a dominant approach to this physics was based on generalizing the SU(2) group of spin rotations to SU(N) or Sp(2N) [1–4]. In the limit $N \to \infty$, models with such enlarged symmetry may be solved exactly by a fully symmetric saddle point of a path integral representation of the partition function, and consequently possess non-magnetic ground states. More recently, it has become possible to study models with SU(N) symmetry for finite $N$ using computational methods. In particular, models describing SU(N) fermions at half filling were shown to host a variety of non-magnetic states [5–10], indicating that features of the $N \to \infty$ solutions survive to $N$ of order one.

Interest in models of this type has also been stimulated by their possible experimental realization in cold atoms [11,12], in materials with orbital degeneracy [13,14], and, more recently, in moiré superlattices with valley degeneracy [15–18]. Here we investigate in particular the SU(4) antiferromagnet in the self-conjugate six dimensional representation. This is the rep-

resentation corresponding to two electrons distributed amongst four degenerate spin/orbital states on each site. This representation thus occurs naturally in systems with spin and a two-fold orbital or valley degeneracy. Like the familiar S=1/2 SU(2) spins, pairs of spins in this representation may form a singlet "valence bond", so that there is a natural "dimer" picture upon which non-magnetic states may be based.

Indeed, it has been shown by Rokhsar [4] that at $N = \infty$, dimerized states (i.e. products of singlet bonds) are the ground states in the self-conjugate representation for a very wide class of lattices and exchange interactions (including almost all those of physical interest). For $N$ large but finite, it is therefore expected that a quantum dimer model [19–23], which describes the projection of the Hamiltonian to the singlet subspace, should capture the physics of the problem. A dimer model consists of a Hilbert space in which dimers, which stand in for singlets, realize a covering of the lattice, with each site covered by one, and only one dimer. Such models are known to predict quantum spin liquids and various valence bond solid orders, depending on the exact model, dimensionality, and lattice. Notably, the simplest quantum dimer model on the triangular lattice was argued to possess a $\mathbb{Z}_2$ spin liquid ground state [22].

With these motivations, we study the aforementioned SU(4) model on the triangular lattice both analytically and numerically. In contrast to the previous numerical works mentioned above, which focused on bipartite lattices, the triangular lattice considered here does not to our knowledge admit a sign-free Monte Carlo approach for generic parameters (though for a specific choice of parameters a sign-free algorithm exists [24]). Hence we attack the problem differently, using the density matrix renormalization group (DMRG), exact diagonalization, and an analytic dimer expansion. First, we describe a reformulation which takes advantage of the fact that SU(4) is a double cover of SO(6), through a mapping of the self-conjugate representation of SU(4) to the vector representation of SO(6), which we define. We determine the classical ground states of the model, and find that they generalize the three-sublattice coplanar states of the S=1/2 Heisenberg model on the triangular lattice. However, carrying out a numerical study of the model using the DMRG method, we argue that this order is absent in the quantum limit. We note that this result is in agreement with a recent pseudo-fermion functional renormalization group study [25] of the same model. We then extend the "overlap expansion" approach – originally developed by Rokhsar and Kivelson [23] to obtain the quantum dimer model Hamiltonian from the SU(2) spin Hamiltonian – to derive an analytic overlap expansion for the quantum dimer model relevant to the SU(4) case. We show that the parameter for the overlap expansion is $x = 1/6$ in the SU(4) case, which should be compared to $x = 1/2$ for SU(2) spins: hence the expansion is expected to be much more accurate for the present problem. Our exact diagonalization (ED) study of the effective dimer model suggests that the ground state is a twelve-site valence bond solid (VBS), although larger system sizes are required to conclusively rule out a proximate quantum spin liquid state. We remark that the related model of Ref. [24] shows such VBS order when generalized from SO(6) to SO(N) with N>12, which provides some further indications for this VBS state in this family of model Hamiltonians.

In addition, we study a generalization of the model which breaks SU(4) symmetry down to SU(2)×SU(2), by including an additional "atomic" Hund's coupling $J_H$ on the sites. In the limit of large $J_H$, the model reduces to the SU(2) symmetric spin S=1 Heisenberg Hamiltonian, and the ground state has long-range three sublattice order. The generalized model thus exhibits a quantum phase transition from a paramagnetic to antiferromagnetically ordered state at zero temperature by varying $J_H$. We present signs for this transition in numerics.

## 2 Model

### 2.1 From Hubbard to Heisenberg

In the limit of strong interactions, electrons localize and the appropriate description of their physics becomes that of their spins, localized at lattice sites. Relevant models may be derived from Hubbard models. Here we proceed with this approach and consider two electrons per site hopping on the triangular lattice with hopping parameter $t$ and subjected to an on-site Hubbard interaction $U$ as well as a Hund's coupling $J_H$, which tends to enforce an alignment of the spin degrees of freedom and thus breaks SU(4) symmetry. More precisely, we consider the following Hamiltonian:

$$H = -t \sum_{\langle ij \rangle} \sum_{a=1}^{4} (c_{ia}^{\dagger} c_{ja} + \text{h.c.}) + U \sum_i n_i(n_i - 2) - J_H \sum_i \left( c_i^{\dagger} \boldsymbol{\sigma} c_i \right)^2, \tag{1}$$

where $c_{i,a}^{\dagger}, c_{i,a}$ are flavor $a$ electron creation and annihilation operators at site $i$, $n_i = \sum_{a=1}^{4} c_{i,a}^{\dagger} c_{i,a}$. In the final term, we used a notation based on the physical origin of the four electronic states from the spin-1/2 of the electron $s^z = \pm 1/2$ and a two-fold orbital degeneracy $\tau^z = \pm 1/2$. We may associate $a = 1, 2, 3, 4$ (note the use of sans-serif font for this purpose) to the states with $(2s^z, 2\tau^z) = (1, 1), (-1, 1), (1, -1), (-1, -1)$, respectively. The $\sigma^{\mu} = \sigma^{\mu} \otimes \mathbb{I}_2$ ($\mu = x, y, z$) Pauli matrices act in spin space. We specialize to the case of filling $n_i = 2$, and carry out the standard degenerate perturbation theory in $t/U$ to derive an effective "spin" model. A basis for the states with $n_i = 2$ on a single site is:

$$\begin{aligned}
|1\rangle = |12\rangle, \qquad |2\rangle = |13\rangle, \qquad |3\rangle = |14\rangle, \\
|4\rangle = |23\rangle, \qquad |5\rangle = |24\rangle, \qquad |6\rangle = |34\rangle,
\end{aligned} \tag{2}$$

where, on the right-hand-side of the equations, $|ab\rangle = c_a^{\dagger} c_b^{\dagger} |0\rangle$, where $|0\rangle$ is the vacuum.

We start by analyzing the SU(4) symmetric model, i.e. we take $J_H = 0$, and return to the effects of a finite $J_H$ later in Sec. 3.3.2. At second order in small $t/U$, Eq. (1) becomes

$$\hat{H} = J \sum_{\langle ij \rangle} \left( \sum_{a,b=1}^{4} \hat{T}_i^{ab} \hat{T}_j^{ba} - \hat{\text{Id}}_6 \right), \tag{3}$$

with $J \sim t^2/U$ and $\hat{T}^{ab} = P_6 c_a^{\dagger} c_b P_6$, with $P_6$ the projection operator onto the six-dimensional vector space defined above. The $\hat{T}^{ab}$ are 6×6 matrices which are related to the generators of SU(4). In Eq. (3) we extracted a constant which sets the zero of energy at a convenient value. To see this, we bring out the analogy to SU(2) spins by extracting the trace from the $\hat{T}^{ab}$ matrices:

$$\tilde{T}^{ab} = \hat{T}^{ab} - \frac{1}{2} \delta^{ab} \hat{\text{Id}}_6. \tag{4}$$

With this definition $\text{Tr} \, \tilde{T}^{ab} = 0$ (note that the trace here is over the 6-dimensional SU(4) space). The Hamiltonian in Eq. (3) can now be written as

$$\hat{H} = J \sum_{\langle ij \rangle} \sum_{a,b=1}^{4} \tilde{T}_i^{ab} \tilde{T}_j^{ba}. \tag{5}$$

One can also check that $\sum_a \tilde{T}^{aa} = 0$, so that there are clearly only 15 such independent SU(4) matrices, which comprise a basis for the generators of SU(4) in the 6-dimensional representation.

Now, regardless of the precise microscopic Hamiltonian $H$, we may consider a spin model, determined on the basis of, and constrained by symmetry. To proceed to the derivation of the most general SU(4) model, it is useful to make use of the following.

## 2.2 Map to the vector representation of SO(6)

SU(4) is a double cover of SO(6) and there exists a convenient map (which is faithful) from the six-dimensional representation of SU(4) to the fundamental (vector) representation of SO(6) [26]. By using the following basis, where each basis state transforms under the vector representation of SO(6), i.e. $O : |\hat{n}\rangle \mapsto O|\hat{n}\rangle$, where $O$ is an SO(6) matrix,

$$
\begin{aligned}
|\hat{1}\rangle &= \frac{1}{\sqrt{2}}(|2\rangle - |5\rangle), & |\hat{2}\rangle &= \frac{-i}{\sqrt{2}}(|2\rangle + |5\rangle), \\
|\hat{3}\rangle &= \frac{1}{\sqrt{2}}(|3\rangle + |4\rangle), & |\hat{4}\rangle &= \frac{-i}{\sqrt{2}}(|3\rangle - |4\rangle), \\
|\hat{5}\rangle &= \frac{1}{\sqrt{2}}(|1\rangle + |6\rangle), & |\hat{6}\rangle &= \frac{-i}{\sqrt{2}}(|1\rangle - |6\rangle),
\end{aligned}
\tag{6}
$$

it is straightforward to write all the SU(4) invariant two-site operators:

$$
\hat{\mathrm{Id}}_{ij} = \sum_{n,m=1}^{6} (|\hat{n}\rangle\langle\hat{n}|)_i (|\hat{m}\rangle\langle\hat{m}|)_j,
\tag{7}
$$

$$
6\hat{P}_{ij} = \hat{Q}_{ij} = \sum_{n,m=1}^{6} (|\hat{n}\rangle\langle\hat{m}|)_i (|\hat{n}\rangle\langle\hat{m}|)_j,
\tag{8}
$$

$$
\hat{\Pi}_{ij} = \sum_{n,m=1}^{6} (|\hat{n}\rangle\langle\hat{m}|)_i (|\hat{m}\rangle\langle\hat{n}|)_j.
\tag{9}
$$

Here $\hat{P}_{ij}$ is the singlet projector over sites $ij$, where a (normalized) singlet over sites $ij$ is written

$$
|s\rangle_{ij} = \frac{1}{\sqrt{6}} \sum_{n=1}^{6} |\hat{n}\hat{n}\rangle_{ij},
\tag{10}
$$

while $\hat{\Pi}_{ij}$ is the permutation operator over sites $ij$ and $\hat{\mathrm{Id}}_{ij}$ is the identity. Then the general SO(6) invariant Hamiltonian with nearest-neighbor interactions is a sum of these terms:

$$
\hat{H}_{\mathrm{gen}} = \sum_{\langle ij \rangle} \left( \alpha \hat{Q}_{ij} + \beta \hat{\Pi}_{ij} + \gamma \hat{\mathrm{Id}}_{ij} \right),
\tag{11}
$$

for $\alpha, \beta, \gamma \in \mathbb{R}$. The "Heisenberg" model Eq. (3) is realized for $-\alpha = \beta = J$, $\gamma = 0$. One can readily check then that for two sites with the SU(4) (or SO(6)) singlet in Eq. (10),

$$
\hat{H}_{ij}|s\rangle_{ij} = J\left( -\hat{Q}_{ij} + \hat{\Pi}_{ij} \right) |s\rangle_{ij} = -5J|s\rangle_{ij}.
\tag{12}
$$

In this SO(6) basis, we may also define the symmetric $\hat{S}^{mn}$ and antisymmetric operators $\hat{\mathcal{A}}^{mn}$, as well as the Hermitian (and still traceless), versions of the latter, $\hat{A}^{mn} = i\hat{\mathcal{A}}^{mn}$

$$
\hat{S}^{mn} = \frac{1}{\sqrt{2}}(|\hat{m}\rangle\langle\hat{n}| + |\hat{n}\rangle\langle\hat{m}|),
\tag{13}
$$

$$
\hat{\mathcal{A}}^{mn} = \frac{1}{\sqrt{2}}(|\hat{m}\rangle\langle\hat{n}| - |\hat{n}\rangle\langle\hat{m}|).
\tag{14}
$$

$$
\hat{A}^{mn} = i\hat{\mathcal{A}}^{mn}.
\tag{15}
$$

The $\hat{A}^{mn}$ operators can be considered as the generators of SO(6), and their square, $\mathrm{Tr}[\hat{\mathbf{A}} \cdot \hat{\mathbf{A}}^T] = 5\hat{\mathrm{Id}}_6$ is the quadratic Casimir operator, up to a normalization constant. Here

we have *defined* the matrix of operators $\hat{\mathbf{A}}$ such that $(\hat{\mathbf{A}})_{mn} = \hat{A}^{mn}$. Using these operators, the "Heisenberg" Hamiltonian $\hat{H}$ becomes

$$\hat{H} = J \sum_{\langle ij \rangle} \sum_{m,n=1}^{6} \hat{A}_i^{mn} \hat{A}_j^{mn} = J \sum_{\langle ij \rangle} \text{Tr} \, \hat{\mathbf{A}}_i \cdot \hat{\mathbf{A}}_j^T, \tag{16}$$

where the trace, $\cdot$ and transpose operations act on the superscripts of the $\hat{\mathbf{A}}_l$ matrices of $\hat{A}_l^{mn}$ operators.

# 3  Magnetic order

In this section, we first examine the classical ground states of the SU(4) model, which are "magnetically" ordered, i.e. they break the SU(4) symmetry and have a non-zero expectation value of the "spin" operator matrix $\tilde{T}_i^{ab}$ or $\hat{A}_i^{mn}$ on each site. Having identified the *type* of magnetic order which is most favored, we next describe numerical studies which search for it. We find that this magnetic order is in fact absent, and that the ground state appears to lack any form of SU(4) symmetry breaking, i.e. is non-magnetic.

## 3.1  Classical limit

In order to look for a product ground state we first ask about the definition of the classical limit of SU(4) (SO(6)) spins. Like for SU(2) spins, we should replace, in the Hamiltonian, each of the fifteen SO(6) generators $\hat{A}^{mn}$, which are 6×6 matrices, by a single classical number.

To do so, we interpret the classical limit as a variational problem in the subspace of states consisting of direct products of single-site wavefunctions. Within any such state, the expectation value of any product of $\hat{A}^{mn}$ is replaced by a product of expectation values of each $\hat{A}^{mn}$, which are c-numbers, as desired. A general single-site wavefunction is given by $|\psi\rangle_i = \sum_{p=1}^{6} v_p^i |\hat{p}\rangle_i$, where $\mathbf{v}^i$ is a complex six-dimensional unit vector, i.e. $\sum_p |v_p^i|^2 = 1$, so that $|\psi\rangle_i$ is normalized. Upon going to the classical limit,

$$\hat{A}^{mn} \to A_{mn} = \langle \psi | \hat{A}^{mn} | \psi \rangle = \sqrt{2} \text{Im}[v_m v_n^*]. \tag{17}$$

The matrix A is now an antisymmetric $6 \times 6$ matrix of scalar matrix elements $A_{mn}$, and the Heisenberg Hamiltonian becomes

$$\hat{H} = J \sum_{\langle ij \rangle} \sum_{m,n=1}^{6} \hat{A}_i^{mn} \hat{A}_j^{mn} \to J \sum_{\langle ij \rangle} \sum_{m,n=1}^{6} A_{mn}^i A_{mn}^j = J \sum_{\langle ij \rangle} \text{Tr} \, A^i (A^j)^T. \tag{18}$$

Note that, while each *operator* $\hat{A}^{mn}$ for fixed $m, n$ is Hermitian, the matrix A is real and anti-symmetric. Moreover, while $\sum_{m,n=1}^{6} \hat{A}^{mn} \hat{A}^{mn} = 5 \hat{\text{Id}}_6$, one can show that $0 \le \text{Tr} AA^T \le 1$ (see Appendix A). In solving a classical SO(6) in this representation model, one should find matrices A which verify the above constraints (much like SU(2) S=1/2 (resp. S=1) classical spins are described by a three-dimensional vector with unit norm $|\mathbf{S}| = 1$ (resp. with $0 \le |\mathbf{S}| \le 1$).

## 3.2  Product variational states

We now specialize to the triangular lattice and nearest-neighbor "Heisenberg" Hamiltonian, and look for the ground state of the corresponding classical model. The SU(2)-invariant spin-1/2 model on the triangular lattice is one of the best-studied models of frustrated magnetism. Its ground states are the so-called 120°-ordered states. They triple the unit cell and each elementary triangular unit is such that the spins point at 120 degrees of one another.

We may rewrite the classical version of the Heisenberg Hamiltonian in Eq. (18), as

$$\mathsf{H} = \frac{J}{4} \sum_{t \; triangle} \left[ \mathrm{Tr}\left( \sum_{i \in t} \mathsf{A}_i \right)\left( \sum_{i \in t} \mathsf{A}_i \right)^T - \mathrm{Tr}\left( \sum_{i \in t} \mathsf{A}_i \mathsf{A}_i^T \right) \right], \tag{19}$$

where the sum runs over all unit triangles of the triangular lattice. The energy is clearly minimized when the first term in the square brackets vanishes and the second one is maximized. The magnitude of the second term is maximized when the upper bound on $\mathrm{Tr}\mathsf{A}_i\mathsf{A}_i^T$ is saturated for all $i \in t$. As shown in Appendix A, the upper bound is saturated when the complex vector $\mathbf{v}$ describing the single-site wavefunction is given by $\mathbf{v} = (\mathbf{x} + i\mathbf{y})/\sqrt{2}$, with $\mathbf{x}, \mathbf{y}$ real, six-dimensional orthogonal unit vectors, i.e. $|\mathbf{x}| = |\mathbf{y}| = 1$ and $\mathbf{x} \cdot \mathbf{y} = 0$. The first term vanishes for three-sublattice states that satisfy $\mathsf{A}_1 + \mathsf{A}_2 + \mathsf{A}_3 = 0_6$.

To minimize the first term in the square brackets in Eq. (19), and simultaneously maximize the magnitude of the second term, we can choose

$$\mathbf{v}_{l=1,2,3} = \mathbf{v}\left(\tfrac{2\pi l}{3}\right), \qquad \text{with} \qquad \mathbf{v}(\theta) = \frac{1}{\sqrt{2}}\left(\mathbf{x} + i(\cos\theta\,\mathbf{y} + \sin\theta\,\mathbf{z})\right), \tag{20}$$

corresponding to

$$\mathsf{A}_{l=1,2,3} = \mathsf{A}\left(\tfrac{2\pi l}{3}\right), \quad \text{with} \quad \mathsf{A}(\theta) = \frac{1}{\sqrt{2}}\left[(\cos\theta\,\mathbf{y} + \sin\theta\,\mathbf{z})\mathbf{x}^T - \mathbf{x}(\cos\theta\,\mathbf{y} + \sin\theta\,\mathbf{z})^T\right], \tag{21}$$

where $\mathbf{x}, \mathbf{y}, \mathbf{z}$ are three orthonormal unit vectors such that $\mathbf{x} \cdot \mathbf{y} = \mathbf{x} \cdot \mathbf{z} = \mathbf{y} \cdot \mathbf{z} = 0$.

For $\mathbf{x}, \mathbf{y}, \mathbf{z}$ along each of the first three basis vectors of $\mathbb{R}^6$, we get for example $\mathsf{A}(\theta) = \cos\theta\,\mathcal{A}^{21} + \sin\theta\,\mathcal{A}^{31}$, where $(\mathcal{A}^{\mu\nu})_{pq} = \frac{1}{\sqrt{2}}(\delta_{\mu p}\delta_{\nu q} - \delta_{\mu q}\delta_{\nu p})$. Note that once the spins on two nearest-neighbor sites are fixed, the remainder are fully determined by the condition $\mathsf{A}_1 + \mathsf{A}_2 + \mathsf{A}_3 = 0_6$, which can be successively applied to the spins on triangles sharing two of the sites which have already been fixed, to cover the entire lattice. This implies that all classical ground states are of the three-sublattice type.

## 3.3 Numerical analysis using DMRG

To probe the presence of magnetic order in the system we study the model numerically, using DMRG [27, 28]. To this end, we consider finite cylinders in a geometry that allows for the formation of a 120°-ordered state. Denoting the basis vectors of the triangular lattice by $\vec{a}_1 = (1, 0), \vec{a}_2 = (1/2, \sqrt{3}/2)$, we consider cylinders such that the sites of the lattice modulo $\vec{R} = N_y\vec{a}_2$ are identified, and $N_y$ is a multiple of three. This geometry is depicted in Fig. 1(c,d). Due to the large single-site Hilbert space dimension in this problem we are limited to narrow cylinders with $N_y = 3$ and $N_y = 6$, and we only study very short cylinders for the latter. We note that a different geometry, namely one in which lattice sites modulo $\vec{R}' = N_y(\vec{a}_2 - \vec{a}_1/2)$ are identified, is also compatible with a 120°-ordered state for $N_y = 4$. However, in this case, we find indications that the system behaves as a quasi-1D system with localized modes at the ends of the cylinder. We thus leave out these results from the discussion of the 2D limit presented here.

In the following, we first discuss the flavor gap in the system, and show that it remains finite, suggesting the absence of a low-energy Goldstone mode that would be expected if the system formed a magnetically ordered state. We then probe the presence of long range order in the ground state by looking at the static response of the system to polarizing fields applied at its boundary. We show that the expectation value of the magnetization decays rapidly away from the boundary in the presence of SU(4) symmetry, implying a lack of long range order.

Our DMRG simulations were performed using the ITensor library [29].

### 3.3.1 Flavor gap

We calculate the flavor gap only for cylinders of width $N_y = 3$, as extracting the gap requires finite length scaling, and we are limited to very short systems for cylinders of width $N_y = 6$, as mentioned above. Note that for $N_y = 3$, the length of the system $N_x$ has to be even to allow for an $SU(4)$-singlet ground state.

Similarly to the conservation of $s^z$ which is often used when studying SU(2) spins, we can employ the conservation of three U(1) quantum numbers for the SU(4) case: $t_3 \equiv n_1 - n_2$, $t_8 \equiv n_1 + n_2 - 2n_3$, and $t_{15} \equiv n_1 + n_2 + n_3 - 3n_4$, where $n_{a=1,..,4}$ denote the occupations of the four flavors as before. We calculate the gap of a $\Delta_{t_3} = 2$ excitation. To this end, we first obtain the ground state, which we expect to be an $SU(4)$ singlet, and hence lie in the $(t_3, t_8, t_{15}) = (0, 0, 0)$ sector, and then calculate the lowest energy state in the $(t_3, t_8, t_{15}) = (2, 0, 0)$ sector. The latter state belongs to the 15 dimensional irreducible representation of $SU(4)$, as can be verified by calculating the quadratic Casimir operator $\sum_{a,b=1}^{4} \tilde{T}^{ab}\tilde{T}^{ba}$. The resulting energy gap is plotted in Fig. 1(a) as function of inverse system length. Even though we present data for relatively short systems, it is clear that the gap remains finite in the infinite system size limit, and we can estimate it to be larger than 2.5$J$. The maximal bond dimension in our simulations was $M = 4000$, resulting in a truncation error of $10^{-5}$ ($10^{-4}$) for the largest system size in the $t_3 = 0$ ($t_3 = 2$) sector.

### 3.3.2 Probing long range magnetic order in the ground state

For the analysis of long range magnetic order in the ground state, it is instructive to consider the effect of $SU(4)$ symmetry breaking by a Hund's coupling term as introduced in Eq. (1). More specifically, the Hamiltonian we consider is

$$\hat{H} = J \sum_{\langle i,j \rangle} \sum_{a,b} \tilde{T}_i^{ab} \tilde{T}_j^{ba} - J_H \sum_i \hat{\mathbf{S}}_i^2. \tag{22}$$

As was mentioned previously, a finite $J_H > 0$ breaks the $SU(4)$ symmetry down to $SU(2) \times SU(2)$, pairing the two electrons on each site into a spin-triplet state. Using the definitions Eq. (2,6), the projection on the on-site spin-triplet and spin-singlet subspaces is given by

$$\hat{\mathcal{P}}_{i,S=1} = \sum_{n=1}^{3} (|\hat{n}\rangle\langle\hat{n}|)_i, \qquad \hat{\mathcal{P}}_{i,S=0} = \sum_{n=4}^{6} (|\hat{n}\rangle\langle\hat{n}|)_i, \tag{23}$$

so that the Hund's coupling term can be written simply as $\hat{\mathbf{S}}_i^2 = \sqrt{2}\hat{\mathcal{P}}_{i,S=1}$.

In the large $J_H$ limit, the spin model Eq. (22) reduces to an SU(2) spin-1 Heisenberg model, $H = J \sum_{\langle ij \rangle} \mathbf{S}_i \cdot \mathbf{S}_j$, where $S_i^\mu$ ($\mu = x, y, z$) are $S = 1$ operators (see Appendix B for further details). The latter is known to form a 120°-ordered state on the triangular lattice [30]. Below, we study the model in Eq. (22) as $J_H$ is increased from $J_H = 0$ (the SU(4) symmetric point), where a three-sublattice order is predicted by our classical analysis, to a large $J_H \gg J$, where the 120° order is known to form also in the quantum limit.

Once $SU(4)$ symmetry is broken down to SU(2)×SU(2), only two U(1) quantum numbers are conserved: the $z$ components of the spin and valley degrees of freedom, namely $2s^z = n_1 + n_3 - n_2 - n_4$ and $2\tau^z = n_1 + n_2 - n_3 - n_4$. Employing the conservation of these two quantum numbers, we now look for the ground state in the sector $(s^z, \tau^z) = (0, 0)$. Once again, we consider finite cylinders of geometry and size compatible with the 3-sublattice order of the 120° state. In particular, we consider cylinders of width $N_y = 3$ and $N_y = 6$ in the same geometry as before.

To facilitate the formation of a long range ordered state, we follow the approach introduced in Ref. [31] and apply pinning fields at the boundaries of the cylinder. We then calculate the

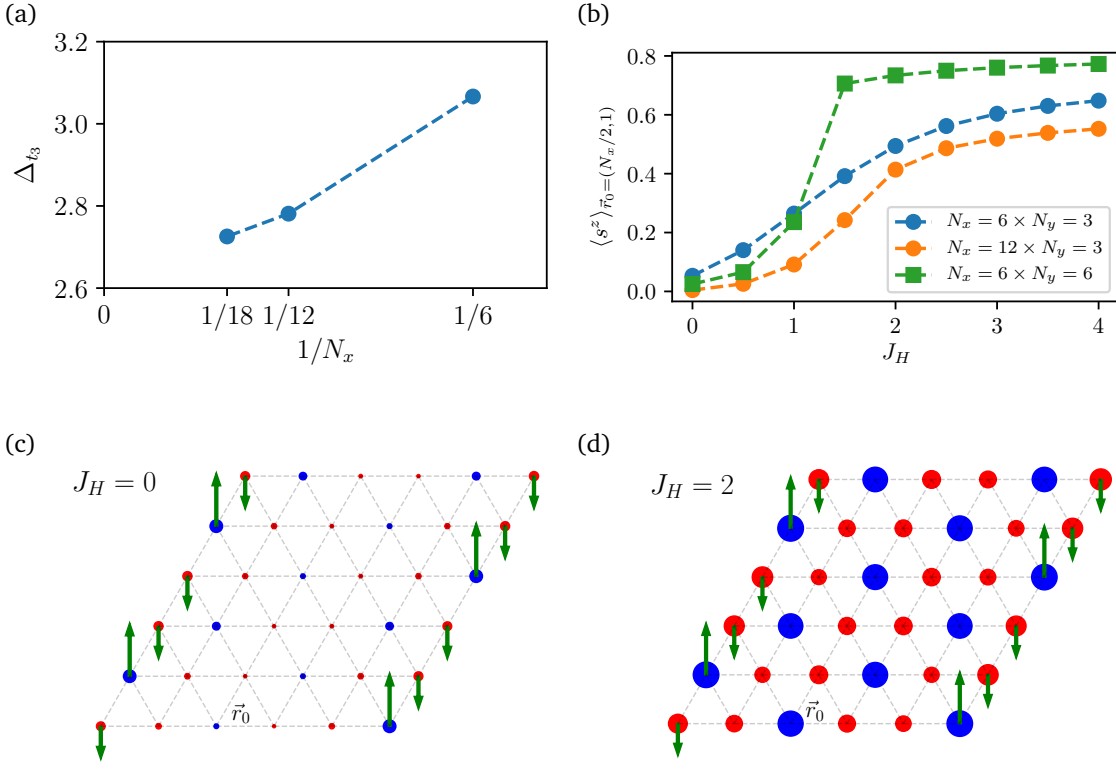

Figure 1: Here we set $J = 1$. (a) Flavor gap, $\Delta_{t_3}$ for $J_H = 0$, as function of inverse system size, obtained for cylinders of width $N_y = 3$. (b) Magnetization on a site in the middle of the system (at position $\vec{r}_0 = (x_0, y_0)$ with $x_0 = N_x/2, y_0 = 1$), as function of the Hund's coupling $J_H$ for different system sizes. Pinning fields are applied along the $z$ axis, on the sites at the boundaries of the cylinder as indicated by the green arrows in (c) and (d), where the local magnetization in a $N_x = 6 \times N_y = 6$ cylinder is shown for $J_H = 0$ and $J_H = 2$ respectively. The area of the circles is proportional to the expectation value of $s^z$ with blue (red) corresponding to a positive (negative) value.

expectation value of the spin component parallel to the field in the bulk, far from the boundary for different ratios of the length of the cylinder to its circumference. A complementary analysis, where we calculate the spin-spin correlations in the absence of pinning fields is presented in Appendix C.1 and gives similar results. To retain the conservation of $s^z$, we apply the pinning fields only along the $s^z$ axis. More specifically, the field applied is $-s^z$ on the A sublattice, and $+s^z/2$ on the B and C sublattices as depicted in Fig. 1(c,d). Note that in the SU(4)-symmetric case this corresponds to a field along $A = \mathcal{A}^{21}$ (see Sec. 3.2).

The expectation value of $s^z$ on a site in the middle of the system, as function of $J_H$, for different system sizes is shown in Fig. 1(b). The expectation value of $s^z$ remains small close to $J_H = 0$, even when the ratio of the length of the cylinder to its circumference is unity, suggesting the absence of magnetic order in this case. As $J_H$ is increased, a finite expectation value develops as expected. The range of system sizes accessible by our simulations is not large enough to perform finite size scaling, but a relatively sharp increase in the magnetization around $J_H/J \simeq 1$ suggests a phase transition occurs in the vicinity of this value. In Figs. 1(c,d) we plot the expectation values of $s^z$ on all the sites of a $6 \times 6$ cylinder, for $J_H = 0$

and $J_H/J = 2$ respectively. While in the former case, the magnetization decays rapidly away from the boundary where the pinning fields are applied, in the latter case the magnetization is finite and uniform across the system.

In these simulations the maximal bond dimension for cylinders of width $N_y = 3$ was $M = 2000$, resulting in a truncation error smaller than $5 \cdot 10^{-5}$. For cylinders of width $N_y = 6$ the maximal bond dimension was $M = 8000$ for $J_H = 0$ and $M = 4000$ for $J_H > 0$, resulting in a truncation error of $\sim 2 \cdot 10^{-3}$ for values of $J_H/J < 1.5$ at which no long-range ordering is observed, and a truncation error of $\sim 5 \cdot 10^{-4}$ or smaller for $J_H/J \geq 1.5$ at which a 120° order develops.

To summarize, our numerical study suggests that the $SU(4)$-symmetric Heisenberg model does not have magnetic long-range order. A transition into a 120°-ordered state can be driven by a Hund's coupling term which breaks $SU(4)$ symmetry.

# 4 Singlet projection

The short-range nature of the spin correlations observed in DMRG motivates an approach focusing on SU(4) singlets. As we saw explicitly in Eq. (10), the six-dimensional representation of SU(4) considered in this work allows for the formation of a singlet on a pair of sites. Hence we can build many singlet states for the entire system by partitioning the sites into pairs, and placing each pair of corresponding spins into a singlet state. Following the pioneering work of Rokhsar and Kivelson [23] who considered the projection of the usual SU(2) Heisenberg model (in the S=1/2 representation) to a nearest-neighbor singlet manifold, we study the projection of the SU(4) Heisenberg model onto the subspace of nearest-neighbor SU(4) singlet "dimer" coverings of the lattice.

In this section, we start with a simple analytic comparison between energies of the singlet states and those of the classical ones discussed earlier, showing that the dimer states are superior in a variational sense. Then we provide further numerical justification for the projection to the singlet subspace. We next discuss the projection of the SU(4) Hamiltonian to the nearest-neighbor singlet coverings subspace and derive an effective dimer model. Finally we study the resulting dimer model using exact diagonalization.

## 4.1 Crude estimate of energy competition between singlet and ordered states

We first estimate the energy of such a singlet state, and compare to that of an ordered state. The optimal ordered product states were found in Sec. 3.2. They comprise 3-sublattice ordered states which spontaneously break SU(4) symmetry analogously to the 120° ordered states for classical SU(2) Heisenberg spins. In those states, the energy per bond is the same for all bonds and is equal to

$$\langle H_{ij} \rangle = J \operatorname{Tr} A^i (A^j)^T = E_{\text{bond}}^{\text{class}} = -\frac{1}{2} J. \tag{24}$$

Hence

$$E^{\text{class}} = -\frac{1}{2} J N_{\text{bonds}} = -\frac{3}{2} J N_{\text{sites}}. \tag{25}$$

Now we consider a singlet state which is the product of two-site singlet "dimers". Specifically, a singlet covering is given by a partition of the set of $N$ sites $i$ into pairs $C = \{(i_1 j_1), (i_2 j_2), \cdots (i_{N/2}, j_{N/2})\}$, where $(i, j)$ denotes a pair of nearest-neighbor sites. Such a state can be visualized by drawing a dimer – a colored bond – between the pairs of sites $(i_a j_a)$. We define

$$|\mathfrak{C}\rangle = \bigotimes_{(ij) \in C} |s\rangle_{ij}, \tag{26}$$

using normalized singlets $|s\rangle_{ij}$ as in Eq. (10). Note that in contrast to the SU(2) case, in the SO(6) representation the singlet state has a purely positive wavefunction, and is without any sign ambiguity. Thus there is no need to define the directionality of a singlet which is required to determine the sign of the wavefunction in the SU(2) case.

For a crude estimate, we consider the variational energy of a single dimer covering,

$$E_{\mathrm{dimer}} = \langle \mathfrak{C}|H|\mathfrak{C}\rangle = \sum_{\langle ij\rangle}\langle \mathfrak{C}|H_{ij}|\mathfrak{C}\rangle. \tag{27}$$

Unlike for the classical state, all the bond expectation values are not equal. As shown in Eq. (12), the singlet $|s\rangle_{ij}$ is an eigenstate of $H_{ij}$, with energy $-5J$. Hence $\langle \mathfrak{C}|H_{ij}|\mathfrak{C}\rangle = -5J$ for those bonds covered by dimers. For bonds that are not covered by singlets, the two spins on the bond are uncorrelated, and one has $\langle \mathfrak{C}|H_{ij}|\mathfrak{C}\rangle = 0$ for those bonds. Hence the variational energy of the dimer state is $-5J$ per bond *times* the fraction of bonds occupied by singlets, which is $1/6$. Thus, since $N_{\mathrm{bonds}} = 3N_{\mathrm{sites}}$

$$E_{\mathrm{dimer}} = -\frac{5}{6}JN_{\mathrm{bonds}} = -\frac{5}{2}JN_{\mathrm{sites}}. \tag{28}$$

Comparing Eq. (28) and Eq. (25), we see that the dimer state has lower energy. This gives some simple understanding of the avoidance of magnetic order.

It is instructive to compare to the SU(2) case, with spin $S$ spins. In this case for the usual Heisenberg model the classical product ground state with 120° order has $J\langle \boldsymbol{S}_i\cdot\boldsymbol{S}_j\rangle_{\mathrm{class}} = -JS^2/2$, so the classical energy is

$$E_{SU(2)}^{\mathrm{class}} = -\frac{JS^2}{2}N_{\mathrm{bonds}}. \tag{29}$$

For a spin singlet bond, we can write $\boldsymbol{S}_i \cdot \boldsymbol{S}_j = \frac{1}{2}[(\boldsymbol{S}_i+\boldsymbol{S}_j)^2 - \boldsymbol{S}_i^2 - \boldsymbol{S}_j^2]$, so that $\langle J\boldsymbol{S}_i\cdot\boldsymbol{S}_j\rangle_{\mathrm{singlet}} = -JS(S+1)$. Thus the dimer energy is

$$E_{SU(2)}^{\mathrm{dimer}} = -\frac{JS(S+1)}{6}N_{\mathrm{bonds}}. \tag{30}$$

Comparing Eq. (29) and Eq. (30), we see that the energies are *equal* for $S = 1/2$ ($E = -JN_{\mathrm{bonds}}/8$), with the classical state superior for all larger $S$.

In summary the simplest possible variational dimer state of a single singlet covering is already better than a classically ordered state for the SU(4) problem, which is distinctly different from the SU(2) case. In the following sections we will refine the approach to singlet states, and consider superpositions of many terms, each with the form of Eq. (26).

## 4.2 Numerical justifications for the projection onto the singlets subspace

We define a nearest-neighbor singlet subspace as the Hilbert space spanned by superpositions of all nearest-neighbor singlet coverings of the form of Eq. (26). In this subsection we compare the low energy spectrum of the Hamiltonian in the full Hilbert space with that of its projection onto this nearest-neighbor singlet space. To this end we perform a numerical study on systems with size of up to 18 sites, using ED for systems with less than 12 sites, and Matrix Product State (MPS)-based simulations for larger systems, as described in detail in Appendix C.2.

We first compare the flavor gap in the $SU(4)$ spin model with the gap in the projected problem. For cylinder of width $N_y = 3$, the flavor gap was discussed in Sec. 3.3 and estimated to be larger than $2.5J$ for an infinitely long cylinder. The gap obtained for the projected problem is $0.281J$, $0.203J$ for system sizes of $N_x = 4$ and $N_x = 6$ respectively. For cylinders of width $N_y = 4$, in the same geometry, we find the flavor gap to be very weakly dependent on system size already for small system sizes, and larger than $3.8J$. The gap in the projected problem is

Table 1: Wavefunction overlaps between the ground state of the $SU(4)$ spin model and the ground state of the Hamiltonian projected onto the subspace of nearest-neighbor singlet coverings for different system sizes ($N_x \times N_y$). "OBC" indicates open boundary conditions along both $x$ and $y$, while "Cylinder" indicates periodic boundary conditions along $y$ and open boundary conditions along $x$. The error indicated in brackets is estimated from the DMRG truncation error for the ground state of the spin model. Values for which no error is indicated were obtained using ED.

|  | $2 \times 2$ | $2 \times 3$ | $4 \times 3$ | $6 \times 3$ | $2 \times 4$ | $3 \times 4$ | $4 \times 4$ |
|---|---|---|---|---|---|---|---|
| OBC | 0.976 | 0.946 | 0.85(2) | 0.76(2) | 0.921 | 0.85(2) | 0.80(4) |
| Cylinder | - | 0.875 | 0.70(1) | 0.55(1) | 0.918 | 0.87(1) | 0.82(1) |

1.738$J$, 1.724$J$ for system sizes of $N_x = 3$ and $N_x = 4$ respectively. Thus, we find that in both cases the gap of the projected Hamiltonian is smaller than the flavor gap, suggesting that the low energy physics is governed by the singlets.

In addition, we calculate the overlaps between the ground state of the $SU(4)$ spin model and that of the projected Hamiltonian. These are summarized in Table 1 for a number of system sizes and different boundary conditions. We find that the overlaps decrease with increasing system size as expected. However, given the immense reduction in the dimension of the Hilbert space upon the projection, we find surprisingly large overlaps even for systems with $N \simeq 10 - 20$ sites.

### 4.3 Derivation of the effective dimer model

We now turn to the analytic derivation of the projected Hamiltonian. Rokhsar and Kivelson [23] constructed an expansion to express the effective projected Hamiltonian as a sum of local terms of increasing length of dimer re-arrangements. We obtain a similar expansion here for the SU(4) ~ SO(6) case. We follow specifically a reformulation of the expansion by Ralko *et al.* [20].

We seek the best variational state of the form

$$|\psi\rangle = \sum_C \psi_C |\mathfrak{C}\rangle. \tag{31}$$

The wavefunction $\psi_C$ is required to minimize

$$E(\psi) = \frac{\langle \psi | H | \psi \rangle}{\langle \psi | \psi \rangle} = \frac{\psi^\dagger \mathsf{H} \psi}{\psi^\dagger \mathsf{S} \psi}, \tag{32}$$

where

$$\mathsf{H}_{C'C} = \langle \mathfrak{C}' | H | \mathfrak{C} \rangle, \qquad \mathsf{S}_{C'C} = \langle \mathfrak{C}' | \mathfrak{C} \rangle. \tag{33}$$

The minimum of the variational energy is given by the condition $\partial E / \partial \psi^* = 0$. This gives

$$\mathsf{H}\psi = E_0 \mathsf{S}\psi, \tag{34}$$

where $E_0 = \min_\psi E(\psi)$ is the best variational energy. This is a generalized eigenvalue problem for $E_0$. We can convert it to a conventional one by defining $\Psi = \mathsf{S}^{1/2}\psi$, which leads to

$$\mathsf{H}_{\text{eff}}\Psi = E_0 \Psi, \tag{35}$$

with the effective Hamiltonian

$$\mathsf{H}_{\text{eff}} = \mathsf{S}^{-1/2} \mathsf{H} \mathsf{S}^{-1/2}. \tag{36}$$

Therefore the variational ground state energy (and from it ultimately the variational ground state wavefunction) is obtained from the ground state of $H_{\text{eff}}$, which is the desired effective quantum dimer Hamiltonian.

To obtain $H_{\text{eff}}$, we expand both $H$ and $S$ in a series of increasingly small terms, which are related to the number of dimer rearrangements forming "loops". The small parameter of this expansion is the overlap $x$ in the smallest such non-trivial loop: two dimers cyclically permuted on four sites. More generally, the inner product of a sequence of dimers pairing sites $C = \{(i_1 j_1), (i_2 j_2), \cdots (i_{N/2}, j_{N/2})\}$ and $C' = \{(i_1 j_2), (i_2 j_3), \cdots (i_{N/2}, j_1)\}$ is $\langle \mathcal{C}' | \mathcal{C} \rangle = x^{N/2-1}$, with $x = 1/6$. In a full calculation of $H$ and $S$, products of such overlaps appear, resulting in multiple factors of $x$, which determines the order of these terms in the expansion. Details of this quite technical procedure, which we formulate for SU(4) on a general lattice, will be presented in a separate publication. Starting from the general SO(6) invariant Hamiltonian in Eq. (11), carrying out this expansion, and then calculating $H_{\text{eff}}$ consistently to a given order gives the final result for the quantum dimer model Hamiltonian:

$$
\begin{aligned}
H_{\text{eff}} = \sideset{}{'}\sum &-t\Big( |\square\rangle\langle\square| + |\square\rangle\langle\square| \Big) + v\Big( |\square\rangle\langle\square| + |\square\rangle\langle\square| \Big) \\
&- t_{6,a}\Big( |\square\rangle\langle\square| + |\square\rangle\langle\square| + \big|\square\big\rangle\big\langle\square\big| + \big|\square\big\rangle\big\langle\square\big| \Big) \\
&- t_{6,b}\Big( \big|\square\big\rangle\big\langle\square\big| + \big|\square\big\rangle\big\langle\square\big| \Big) \\
&+ u\Big( |\square\rangle\langle\square| + |\square\rangle\langle\square| + |\square\rangle\langle\square| + |\square\rangle\langle\square| \\
&\qquad\quad + \big|\square\big\rangle\big\langle\square\big| + \big|\square\big\rangle\big\langle\square\big| + \big|\square\big\rangle\big\langle\square\big| + \big|\square\big\rangle\big\langle\square\big| \Big) \\
&- t_{8,a}\Big( \big|\square\big\rangle\big\langle\square\big| + \big|\square\big\rangle\big\langle\square\big| \Big) - t_{8,b}\Big( \big|\square\big\rangle\big\langle\square\big| + \big|\square\big\rangle\big\langle\square\big| \Big) \\
&- t_{8,c}\Big( \big|\square\big\rangle\big\langle\square\big| + \big|\square\big\rangle\big\langle\square\big| \Big) - t_{8,d}\Big( \big|\square\big\rangle\big\langle\square\big| + \big|\square\big\rangle\big\langle\square\big| \Big) \\
&- t_{8,e}\Big( \big|\square\big\rangle\big\langle\square\big| + \big|\square\big\rangle\big\langle\square\big| + \big|\square\big\rangle\big\langle\square\big| + \big|\square\big\rangle\big\langle\square\big| \\
&\qquad\quad + \big|\square\big\rangle\big\langle\square\big| + \big|\square\big\rangle\big\langle\square\big| \Big) \\
&- t_{8,f}\Big( |\square\rangle\langle\square| + |\square\rangle\langle\square| + \big|\square\big\rangle\big\langle\square\big| + \big|\square\big\rangle\big\langle\square\big| \\
&\qquad\quad + \big|\square\big\rangle\big\langle\square\big| + \big|\square\big\rangle\big\langle\square\big| \Big) \\
&- t_{8,g}\Big( \big|\square\big\rangle\big\langle\square\big| + \big|\square\big\rangle\big\langle\square\big| + \big|\square\big\rangle\big\langle\square\big| + \big|\square\big\rangle\big\langle\square\big| \Big),
\end{aligned}
\tag{37}
$$

where the prime on the sum indicates a sum over all the symmetry-equivalent plaquettes shown in the bras and kets, throughout the lattice. All the coefficients are given in terms of $\alpha$, $\beta$, and $x$ and are summarized in Table 2.

## 4.4 Numerical study of the dimer model

We now turn to an analysis of the dimer model obtained in the previous section, Eq. (37), and taking $\alpha = -1$, $\beta = 1$, corresponding to the Heisenberg model.

To zeroth order in the expansion parameter $x$, the dimer model obtained is the standard dimer model considered by Rokhsar and Kivelson [23], i.e. $H = -t(|\square\rangle\langle\square| + |\square\rangle\langle\square|) + v(|\square\rangle\langle\square| + |\square\rangle\langle\square|)$, with $v = 0$ and $t = 1$ – for these values the Hamiltonian contains only the "flip" term, consistent with the truncation of Eq. (37) to zeroth order in $x$. Previous studies of this model on the triangular lattice [21, 22, 32, 33] found that the ground state for $v/t = 0$ is a $\sqrt{12} \times \sqrt{12}$ VBS state. At large enough negative $v/t$, the ground state is a columnar ordered

Table 2: Coefficients of the terms in the effective dimer model given in Eq. (37). The numerical value in the last column is calculated for the parameters corresponding to the SU(4) Heisenberg model, namely $\alpha = -1, \beta = 1, x = 1/6$.

|           | expression                                           | numerical value |
|-----------|------------------------------------------------------|-----------------|
| $t$       | $-(2\alpha + \beta) - x(\alpha + 2\beta) + x^2(\alpha + 2\beta)$ | 31/36 |
| $v,\ t_{6,a}$ | $-x(2\alpha + \beta) - x^2(\alpha + 2\beta)$     | 5/36            |
| $t_{6,b}$ | $-3x(\alpha + \beta) - 3x^2(\alpha + \beta)$         | 0               |
| $u$       | $-\frac{1}{2}x^2(2\alpha + \beta)$                   | 1/72            |
| $t_{8,a}$ | $-4\alpha x^2$                                        | 1/9             |
| $t_{8,b}$ | $-\frac{1}{2}x^2(5\alpha + 6\beta)$                  | -1/72           |
| $t_{8,c}$ | $-x^2(4\alpha + 5\beta)$                             | -1/36           |
| $t_{8,d}$ | $-\frac{1}{2}x^2(5\alpha + 4\beta)$                  | 1/72            |
| $t_{8,e}$ | $-x^2(2\alpha + \beta)$                              | 1/36            |
| $t_{8,f}$ | $-\frac{1}{2}x^2(2\alpha + \beta)$                  | 1/72            |
| $t_{8,g}$ | $-2x^2(\alpha + \beta)$                              | 0               |

state, while for positive $v/t$, first a phase transition into the $\mathbb{Z}_2$ RVB spin liquid phase occurs at $v/t \simeq 0.83$, followed by a transition at $v/t = 1$ into a staggered ordered phase.

### 4.4.1 Geometry and sectors

To understand how higher order terms in the expansion affect the ground state, we now study the model in Eq. (37) numerically, using ED. We consider systems with periodic boundary conditions along both directions (i.e. systems on a torus), and focus on two types of clusters which keep all the symmetries of the infinite lattice, following Ref. [21]. Denoting the basis vectors of the triangular lattice by $\vec{a}_1 = (1, 0)$, $\vec{a}_2 = (1/2, \sqrt{3}/2)$, these clusters are defined by identifying the sites of the lattice modulo the vectors $(\vec{R}_1, \vec{R}_2)$, where $(\vec{R}_1, \vec{R}_2) = m(\vec{a}_1, \vec{a}_2)$ for clusters of type A, and $(\vec{R}_1, \vec{R}_2) = (m\vec{a}_1 + m\vec{a}_2, -m\vec{a}_1 + 2m\vec{a}_2)$ for clusters of type B. These two types of clusters are shown in Fig. 2. Note that the number of sites in cluster of type A (B) is $m^2$ ($3m^2$).

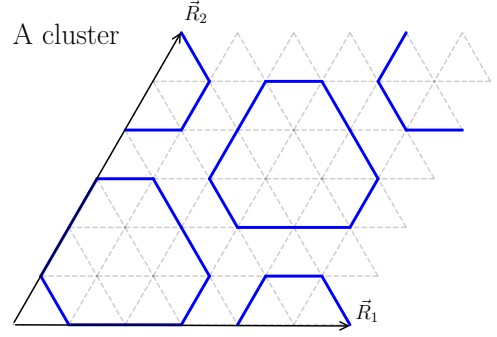
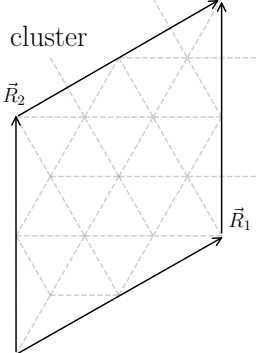

Figure 2: The two types of clusters considered in the numerical study of the dimer model. A schematic representation of the $\sqrt{12} \times \sqrt{12}$ order, depicting the 12-site unit cell is shown for the $6 \times 6$ type A cluster.

On a torus, the Hilbert space of dimer coverings breaks up into four distinct topological sec-

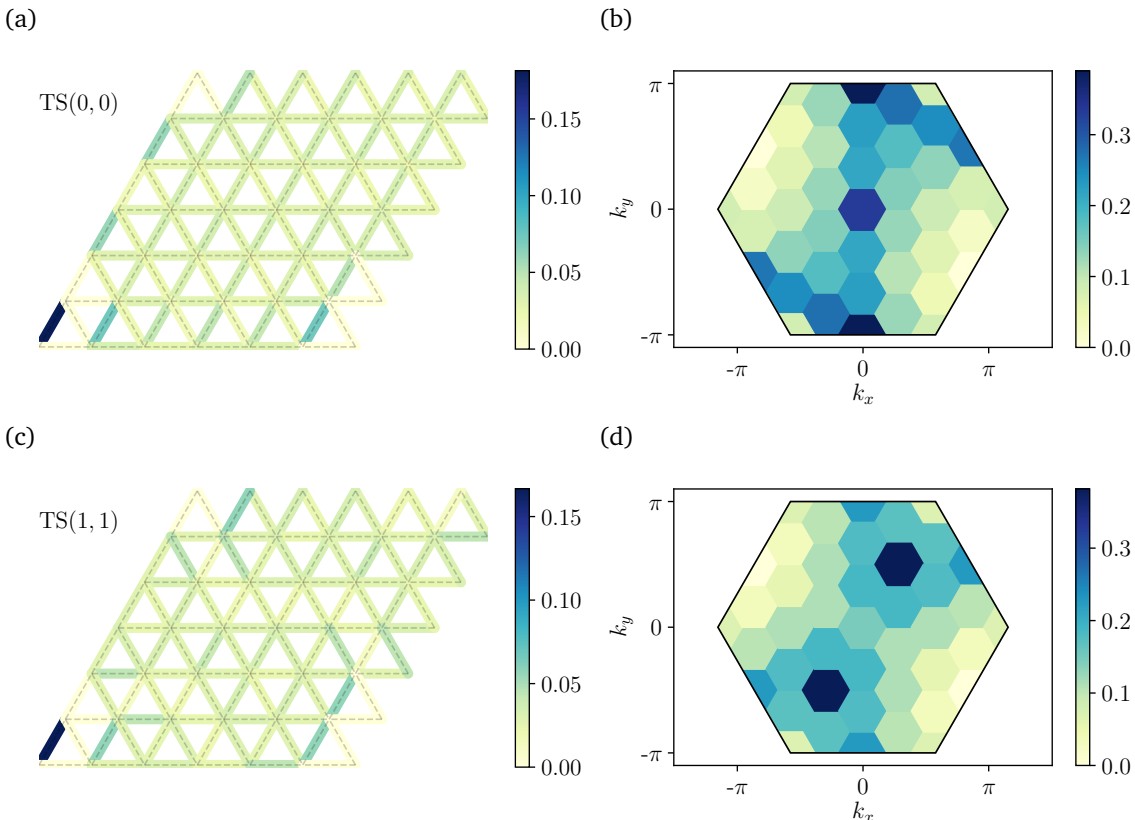

Figure 3: (a,b) Dimer-dimer correlations $\langle b_1 b_i \rangle$ (where $b_i = 1$ if the $i$th bond is occupied by a dimer and $b_i = 0$ otherwise) in the lowest energy state in the topological sectors $TS(0,0)$ and $TS(1,1)$ respectively of the extended dimer model $H_2$ on a $6 \times 6$ lattice obtained using ED. (c,d) Fourier transform of $\langle b_1 b_i \rangle - \langle b_1 \rangle \langle b_i \rangle$ for the vertical bonds in the two topological sectors respectively.

tors defined by the parities of the number of dimers intersected by closed loops winding around the torus along the two axes. We will denote these sectors by $TS(p_x, p_y)$, with $p_x, p_y = 0(1)$ for even (odd) parity along $x$ and $y$ respectively. As pointed out in Ref. [21], on a cluster with $C_6$ symmetry, three of these topological sectors are always degenerate since they can be related by $C_6$ rotations of the lattice. Which three sectors are degenerate depends on the parity of $m/2$, but in order to understand the spectrum of the problem it is enough to consider the two sectors $TS(0,0)$ and $TS(1,1)$, as these two sectors are never related by $C_6$ rotations.

We consider system sizes of up to 36 sites, i.e. clusters of type B with $m = 2$ (12 sites) and clusters of type A with $m = 4$ and $m = 6$ (16 and 36 sites respectively). We note that to allow for a $\sqrt{12} \times \sqrt{12}$ order, the number of sites in the system must be a multiple of 12. In addition, for $m/2$ odd, only the topological sector $TS(1,1)$ can accommodate this ordering without defects (see Fig. 2).

### 4.4.2 Exact diagonalization results

We study the successive approximations obtained by working to increasingly higher order in $x$, denoting by $H_n$ the sum of all terms in the effective Hamiltonian up to and including $O(x^n)$. More explicitly, $H_0$ consists solely of the kinetic term on a plaquette with $t = 1$, while $H_1$ contains in addition the potential energy term $v$ as well as the kinetic terms corresponding to hopping on loops of length six. The values of the coefficients in $H_1$ are given by $t = 5/6$,

and $v = t_{6,a} = 1/6$ (note that $t_{6,b} = 0$). The Hamiltonian $H_2$ contains all the terms in Eq. 37 with the corresponding values given in Table 2. We calculate the lowest energy state in each topological sector of $H_{n=0,1,2}$, for the physical situation $x = 1/6$. The values obtained are summarized in Appendix C.3.1. We find that the correction due to second order terms is indeed small compared to the first order ones.

We then interpolate between the Hamiltonians $H_0$ and $H_2$, calculating the low energy spectrum of $H(\eta) = (1 - \eta)H_0 + \eta H_2$. We find that there are no level crossings in the low energy spectrum, and the ground state remains in the topological sector TS$(1, 1)$ for system sizes which can accommodate the $\sqrt{12} \times \sqrt{12}$ order (see Fig. 5 in Appendix C.3.1). The smooth continuity suggests that $H_0$ and $H_2$ describe the same phase of matter. Furthermore, we find that, in each topological sector, the wavefunction overlap between the lowest energy state of $H_0$ and that of $H_2$ is very close to one, in particular in the topological sector TS$(1, 1)$. More specifically, for the $6 \times 6$ system, the overlaps are 0.88 and 0.97 for TS$(0, 0)$ and TS$(1, 1)$ respectively.

In addition, we compare the dimer-dimer correlations in these states. We find that the correlations in the lowest energy states of $H_2$ become slightly more uniform compared to those in the lowest energy states of $H_0$, but overall display the same features (see Appendix C.3.2). In Fig. 3 we plot the real space dimer-dimer correlations, as well as their Fourier transform, calculated in the lowest energy state of $H_2$ in the two topological sectors for a $6 \times 6$ lattice. As can be clearly seen, for the state in TS$(1, 1)$ sharp peaks at $\vec{k} = \pm(\pi/(2\sqrt{3}), \pi/2)$ are present, suggesting breaking of translational invariance compatible with the formation of a 12-site unit cell. We note that the six-fold rotational symmetry expected in the ground state is broken in Figs. 3(b,d) by the choice of the set of bonds used in the calculation of the dimer-dimer correlation function. The structure factor shown in Figs. 3(b,d) is obtained for the bonds parallel to the lattice basis vector $\vec{a}_2$. When the correlation function is calculated with respect to a set of bonds related by a $\pi/3$ rotation on each site, the peaks in the structure factor appear at momenta related by the corresponding $\pi/3$ rotation.

Although a better finite size scaling analysis is required to make a conclusive statement regarding the nature of the ground state of the dimer model, we believe that these observations – (i) the similarity of the ground state correlations to those of the "standard" dimer model at $v/t = 0$ for small system sizes, and (ii) the smooth evolution of the spectrum upon interpolation between the two models – strongly suggest that the ground state remains a $\sqrt{12} \times \sqrt{12}$ VBS ordered state.

## 5 Conclusion

In this work, we considered SU(4) spins in the six-dimensional (self-conjugate) representation, on the triangular lattice, with nearest-neighbor antiferromagnetic interactions. Our DMRG study suggests that the ground state is non-magnetic, but remains inconclusive as to the exact nature of the ground state. We developed and carried out a dimer expansion, which we argued is capable of capturing the low energy properties of the model. The study of the the associated dimer model led us to conjecture that the ground state of the SU(4) model may be a 12-site valence bond solid (VBS).

As the mapping to the dimer model involves an uncontrolled projection, we do not know how to systematically improve it. Hence, a fully conclusive study should return to the original SU(4) spin model. This, however, remains numerically challenging due to the large on-site Hilbert space dimension. As a first step in this direction, we carried out preliminary calculations in addition to those reported in this paper, using the infinite DMRG (iDMRG) method on width-four cylinders. By choosing appropriate boundary conditions, this geometry is com-

patible with the 12-site VBS order. However, we did not find signatures of this order in our iDMRG simulations. One possible interpretation is that the non-observation of VBS order is simply due to the effects of finite size or finite bond dimension. Another possibility is that the VBS order is truly absent, indicating some type of spin liquid state without broken symmetries. The proximity of a $\mathbb{Z}_2$ spin liquid phase in the effective dimer model suggests this as an intriguing possibility. Regardless, this conundrum highlights the challenges of a direct simulation of the original SU(4) problem.

In the study of the effective dimer model, we focused on the parameters corresponding to the SU(4) Heisenberg model, $\alpha = -1, \beta = 1$. In the future it would be interesting to explore the full phase diagram of the general dimer model derived, understand if it can realize the $\mathbb{Z}_2$ spin liquid phase, and identify the nature of the interactions in terms of the SU(4) spins required for this.

In addition, it would be desirable to study in more detail the evolution of the ground state with increasing $J_H$. If the ground state of the spin model at $J_H = 0$ is indeed a 12-site VBS, and if there is, as suggested by our numerics, a direct transition to a three-sublattice ordered state with increasing $J_H$, then this is a Landau-forbidden quantum phase transition. If this is realized via a continuous quantum critical point, then it must be an example of deconfined quantum criticality. It would be interesting to understand the nature of this critical point and test it in numerics.

# Acknowledgements

A.K. would like to thank Bela Bauer for valuable discussions regarding the numerical studies presented in this work. We thank Chao-Ming Jian for discussions regarding the irreducible representations of SU(4). This research is funded in part by the Gordon and Betty Moore Foundation through Grant GBMF8690 to UCSB to support the work of A.K. Use was made of the computational facilities administered by the Center for Scientific Computing at the CNSI and MRL (an NSF MRSEC; DMR-1720256) and purchased through NSF CNS-1725797. L.S. was supported by the Agence Nationale de la Recherche through Grant ANR-18-ERC2-0003-01 (QUANTEM), and in part by the National Science Foundation under Grant No. NSF PHY-1748958. L.B. was supported by the DOE, Office of Science, Basic Energy Sciences under Award No. DE-FG02-08ER46524.

# A  Classical limit

## A.1  SO(6) formulation

The classical limit is taken by replacing the 15 generators $\hat{A}^{mn}$ by their expectation values in a given state, i.e.:

$$\hat{A}^{mn} \rightarrow \mathsf{A}_{mn} = \langle \psi | \hat{A}^{mn} | \psi \rangle. \tag{38}$$

Since the operators $\hat{A}^{mn}$ are hermitian and satisfy $\hat{A}^{mn} = -\hat{A}^{nm}$, the matrix A is real and anti-symmetric, i.e. $\mathsf{A}^T = -\mathsf{A}$. Note that since A is real, $\operatorname{Tr} \mathsf{A}\mathsf{A}^T > 0$, and therefore $\operatorname{Tr} \mathsf{A}^2 = -\operatorname{Tr} \mathsf{A}\mathsf{A}^T < 0$.

Writing the quantum state explicitly as $|\psi\rangle = \sum_{p=1}^{6} v_p | \hat{p} \rangle$, the matrix elements of A are given by

$$\mathsf{A}_{mn} = \sum_{p,p'} \frac{i v_p^* v_{p'}}{\sqrt{2}} \langle \hat{p} | (|\hat{m}\rangle\langle\hat{n}| - |\hat{n}\rangle\langle\hat{m}|) | \hat{p}' \rangle = \frac{i}{\sqrt{2}} \left( v_m^* v_n - v_n^* v_m \right) = \sqrt{2} \operatorname{Im}[v_n^* v_m], \tag{39}$$

or in matrix notations $A = i(\mathbf{v}^*\mathbf{v}^T - \mathbf{v}\mathbf{v}^\dagger)/\sqrt{2} = \sqrt{2}\,\text{Im}[\mathbf{v}\mathbf{v}^\dagger]$. We next note that

$$\text{Tr}\,AA^T = \sum_{m,n=1}^{6} A_{mn}A_{mn} = 1 - \left|\sum_{n=1}^{6} v_n^2\right|^2 \leq 1. \tag{40}$$

Further let $\mathbf{v} = (\mathbf{x} + i\mathbf{y})/\sqrt{2}$ with $\mathbf{x}$, $\mathbf{y}$ real six-dimensional vectors with unit norm. Then $\text{Tr}\,AA^T = 1 - (\mathbf{x}\cdot\mathbf{y})^2$. It is now easy to see that the upper bound on $\text{Tr}\,AA^T$ is reached when $\mathbf{x} \perp \mathbf{y}$.

In the classical limit the Hamiltonian is given by:

$$H_{\text{Heis}} = J\sum_{\langle ij\rangle}\sum_{m,n=1}^{6} A_{mn}^i A_{mn}^j = J\sum_{\langle ij\rangle}\text{Tr}[A_i A_j^T]. \tag{41}$$

On the triangular lattice we can rewrite:

$$H = \frac{J}{4}\sum_{t\ triangle}\left[\text{Tr}\left(\sum_{i\in t}A_i\right)\left(\sum_{i\in t}A_i\right)^T - \text{Tr}\left(\sum_{i\in t}A_i A_i^T\right)\right]. \tag{42}$$

For antiferromagnetic coupling, $J > 0$, to minimize the energy, we would like the first term to vanish, and the second to be as negative as possible. Let us denote by $\mathbf{x}, \mathbf{y}, \mathbf{z}$ three real, orthogonal, six-dimensional unit vectors, and define

$$\mathbf{v}(\theta) = \frac{1}{\sqrt{2}}\left(\mathbf{y} + i(\cos\theta\,\mathbf{x} + \sin\theta\,\mathbf{z})\right), \tag{43}$$

$$A(\theta) = \sqrt{2}\left(\text{Im}[\mathbf{v}(\theta)]\text{Re}[\mathbf{v}(\theta)]^T - \text{Re}[\mathbf{v}(\theta)]\text{Im}[\mathbf{v}(\theta)]^T\right). \tag{44}$$

Then, the matrices $A_{l=1,2,3} = A\left(\frac{2\pi l}{3}\right)$ satisfy $\sum_{l=1}^{3} A_l = 0$ and $\text{Tr}\,A_l A_l^T = 1$, thus minimizing the energy on a triangle.

Note that choosing $\mathbf{x} = \mathbf{e}_1$, $\mathbf{y} = \mathbf{e}_2$, $\mathbf{z} = \mathbf{e}_3$, with $\mathbf{e}_n$ denoting the unit vector along the $n$th dimension in $\mathbb{R}^6$ we obtain a state $\mathbf{v}(\theta)$ that belongs to the spin-triplet valley-singlet subspace on a given site (see also Appendix B below and in particular Eq. (58) therein). The classical ground state corresponding to the states $\mathbf{v}_{l=1,2,3} = \mathbf{v}(\frac{2\pi l}{3})$ on each triangle of the lattice is then, in this case, exactly the 120° ordered state of the SU(2) spin-ones.

## A.2 SU(4) formulation

Here we derive the classical energy function and constraints using the SU(4) formulation, i.e. starting from the Hamiltonian Eq. (5), and using the basis states on the right-hand-sides of the equalities in Eq. (2).

We start by writing the quantum state on a single site explicitly as $|\psi\rangle = \sum_{a,b=1}^{4}\psi_{ab}c_a^\dagger c_b^\dagger|0\rangle$. The $4\times 4$ matrix $\psi$ must be antisymmetric, $\psi^T = -\psi$, and the normalization constraint $\langle\psi|\psi\rangle = 1$ imposes $\text{Tr}\psi^\dagger\psi = 1/2$. The classical limit is obtained by replacing the 15 generators $\tilde{T}^{ab}$ ($a, b = 1, .., 4$) by their expectation values in a given state $|\psi\rangle$:

$$\tilde{T}^{ab} \to T_{ab} = \langle\psi|\tilde{T}^{ab}|\psi\rangle = 4(\psi^\dagger\psi)_{ab} - \frac{1}{2}\delta_{ab}, \quad \text{i.e.} \quad T = 4\psi^\dagger\psi - \frac{1}{2}\text{Id}. \tag{45}$$

We have in turn $T^\dagger = T$ and $\text{Tr}\,T = 0$.

We now proceed to finding lower and upper bounds on $\text{Tr}\,T^2$, as these will be important for the minimization of the energy. We will show that $0 \leq \text{Tr}\,T^2 \leq 1$. To do so, we consider

the eigenvalues of $\mathsf{T}$. Since $\mathsf{T}$ is hermitian, its eigenvalues $t_n$ are real. Using the inequality $\sum_{n=1}^{N} t_n^2 \geq \frac{1}{N}\left(\sum_{n=1}^{N} t_n\right)^2$, we find the lower bound

$$\mathrm{Tr}\mathsf{T}^2 \geq \frac{1}{4}(\mathrm{Tr}\mathsf{T})^2 = 0, \tag{46}$$

which is saturated for example for $\mathsf{T}_{\mathrm{lower}} = 0_4$, and corresponds to $\psi_{\mathrm{lower}} = \frac{1}{2\sqrt{2}}\sigma^0(i\tau^y)$. The specific form of $\mathsf{T}$ in terms of the square of $\psi$ imposes a stringent upper bound. Indeed, the antisymmetry of $\psi$ makes the latter diagonalizable, and that combined with its even dimension imposes that its eigenvalues come in pairs $\pm y_{1,2}$. In turn the eigenvalues of $\mathsf{T}$ are doubly degenerate and equal to $4|y_{1,2}|^2 - 1/2$. Therefore,

$$\mathrm{Tr}\mathsf{T}^2 = 2\sum_{i=1,2}\left(4|y_i|^2 - 1/2\right)^2 = 32(|y_1|^4 + |y_2|^4) - 1, \tag{47}$$

since $\mathrm{Tr}\mathsf{T} = 8(|y_1|^2 + |y_2|^2) - 2 = 0$. Given $\mathrm{Tr}\psi^\dagger\psi = 2(|y_1|^2 + |y_2|^2) = 1/2$, the maximum of $\mathrm{Tr}\mathsf{T}^2$ is reached for $\{|y_1| = 0, |y_2| = 1/2\}$ or $\{|y_1| = 1/2, |y_2| = 0\}$ and equal to 1. This is achieved for example for $\mathsf{T}_{\mathrm{upper}} = \frac{1}{2}\sigma^0\tau^z$, which corresponds to $\psi_{\mathrm{upper}} = \frac{1}{4}(i\sigma^y)(\tau^0 + \tau^z)$.

In summary, a classical SU(4) "spin" $\mathsf{T}$ satisfies:

$$\mathsf{T}^\dagger = \mathsf{T}, \qquad \mathrm{Tr}\mathsf{T} = 0, \qquad 0 \leq \mathrm{Tr}\mathsf{T}^2 \leq 1. \tag{48}$$

In this formulation, in the classical limit the Hamiltonian is given by:

$$\mathsf{H}_{\mathrm{Heis}} = J\sum_{\langle ij\rangle}\sum_{a,b=1}^{4}\mathsf{T}_{ab}^i\mathsf{T}_{ba}^j = J\sum_{\langle ij\rangle}\mathrm{Tr}[\mathsf{T}_i\mathsf{T}_j]. \tag{49}$$

On the triangular lattice we can rewrite:

$$\mathsf{H} = \frac{J}{4}\sum_{t\,triangle}\left[\mathrm{Tr}\left(\sum_{i\in t}\mathsf{T}_i\right)^2 - \mathrm{Tr}\left(\sum_{i\in t}\mathsf{T}_i^2\right)\right]. \tag{50}$$

For antiferromagnetic coupling, $J > 0$, to minimize the energy, we would like the first term to vanish, and the second to be as negative as possible. Let us denote by $\mathbf{z}, \mathbf{x}$ two real, orthogonal, three-dimensional unit vectors, and define

$$\mathbf{n}(\theta) = \cos\theta\,\mathbf{z} + \sin\theta\,\mathbf{x}, \tag{51}$$

$$\mathsf{T}(\theta) = \frac{1}{2}\sigma^0(\mathbf{n}(\theta)\cdot\boldsymbol{\tau}). \tag{52}$$

Then, the matrices $\mathsf{T}_{l=1,2,3} = \mathsf{T}\left(\frac{2\pi l}{3}\right)$ satisfy $\sum_{l=1}^{3}\mathsf{T}_l = 0$ and $\mathrm{Tr}\,\mathsf{T}_l^2 = 1$, thus minimizing the energy on a triangle. The corresponding state $\psi_l$ can be chosen to be

$$\psi_l = \frac{1}{4}(i\sigma^y)\left(\tau^0 + \mathbf{n}(\theta_l)\cdot\boldsymbol{\tau}\right), \tag{53}$$

where $\theta_l = 2\pi l/3$, so that

$$|\psi_l\rangle = \frac{1}{2}\left[(|13\rangle + |24\rangle) + (|13\rangle - |24\rangle)\cos\theta_l + (|14\rangle + |23\rangle)\sin\theta_l\right]. \tag{54}$$

Indeed, choosing $\mathbf{z} = (0,0,1)$ and $\mathbf{x} = (1,0,0)$, we have $\psi_0 = \frac{1}{2}(i\sigma^y)\otimes\begin{pmatrix}1 & 0\\0 & 0\end{pmatrix} = |13\rangle$, and $\psi_l$ is obtained from $\psi_0$ through the rotation $\psi_l = R_l^T\psi_0 R_l$, with $R_l = \sigma^0 r_l$, where

$$r_l = \exp[\frac{i}{2}\frac{2\pi l}{3}\mathbf{y}\cdot\boldsymbol{\tau}] = \cos\frac{\pi l}{3} + i\mathbf{y}\cdot\boldsymbol{\tau}\sin\frac{\pi l}{3}, \tag{55}$$

where $\mathbf{y} = (0,1,0)$.

### A.3  Mapping between the SO(6) and SU(4) formulations

Here we describe the mapping between the SO(6) and SU(4) formulations and show that the classical ground state obtained in the two formulations is indeed the same state.

The six basis states $|\hat{n}\rangle$ in Eq. (6) correspond to the following $4 \times 4$ antisymmetric matrices $\psi$:

$$
\begin{aligned}
|\hat{1}\rangle \to \psi_1 = \frac{i}{2\sqrt{2}}\sigma^y \otimes \tau^z, &\qquad |\hat{2}\rangle \to \psi_2 = \frac{1}{2\sqrt{2}}\sigma^y \otimes \tau^0, \\
|\hat{3}\rangle \to \psi_3 = \frac{i}{2\sqrt{2}}\sigma^y \otimes \tau^x, &\qquad |\hat{4}\rangle \to \psi_4 = \frac{i}{2\sqrt{2}}\sigma^y \otimes \tau^y, \\
|\hat{5}\rangle \to \psi_5 = \frac{1}{2\sqrt{2}}\sigma^0 \otimes \tau^x, &\qquad |\hat{6}\rangle \to \psi_6 = \frac{1}{2\sqrt{2}}\sigma^z \otimes \tau^z.
\end{aligned}
\tag{56}
$$

Using this mapping one can translate the classical states that optimize the energy on the triangular lattice corresponding to $\mathbf{v}(\theta)$ in Eq. (51) to the corresponding $\psi(\theta)$. More explicitly, for $\mathbf{x} = \mathbf{e}_1$, $\mathbf{y} = \mathbf{e}_2$, $\mathbf{z} = \mathbf{e}_3$ with $\mathbf{e}_n$ denoting the unit vector along the $n$th dimension in $\mathbb{R}^6$ we obtain a state

$$
\psi(\theta) = \frac{1}{4}\sigma^y \left( \tau^0 - \mathbf{n}(\theta) \cdot \boldsymbol{\tau} \right),
\tag{57}
$$

where $\mathbf{n}(\theta) = (\sin\theta, 0, \cos\theta)$ and $\boldsymbol{\tau} = (\tau^x, \tau^y, \tau^z)$. Thus, $\psi(\theta_l)$, with $\theta_l = 2\pi l/3 + \pi$ reproduce the states in Eq. (53) up to an overall phase.

## B  Large Hund's coupling limit

In the large Hund's coupling limit, i.e. $J_H/J \gg 1$, the term $-J_H \sum_i \mathbf{S}_i^2$ in Eq. (22) requires the total spin at each site to be in the $S = 1$ representation of SU(2). The associated vector space is spanned by

$$
\begin{aligned}
|S = 1, s^z = 1\rangle &\quad = \quad |2\rangle &\quad = \quad \frac{1}{\sqrt{2}}(|\hat{1}\rangle + i|\hat{2}\rangle), \\
|S = 1, s^z = 0\rangle &\quad = \quad \frac{1}{\sqrt{2}}(|3\rangle + |4\rangle) &\quad = \quad |\hat{3}\rangle, \\
|S = 1, s^z = -1\rangle &\quad = \quad |5\rangle &\quad = \quad \frac{1}{\sqrt{2}}(-|\hat{1}\rangle + i|\hat{2}\rangle),
\end{aligned}
\tag{58}
$$

and thus the operator $\hat{\mathcal{P}}_{i,S=1} = \sum_{n=1}^{3} |\hat{n}_i\rangle\langle\hat{n}_i|$ projects the state on site $i$ onto the $S = 1$ subspace. Note also that $|S = 1, s^x = 0\rangle = |\hat{1}\rangle$ and $|S = 1, s^y = 0\rangle = |\hat{2}\rangle$, and thus the $S = 1$ spin operators can be written as

$$
S^z = \sqrt{2}\hat{A}^{21}, \qquad S^x = \sqrt{2}\hat{A}^{32}, \qquad S^y = \sqrt{2}\hat{A}^{13}.
\tag{59}
$$

Denoting by $\hat{\mathcal{P}}_{S=1} = \prod_i \hat{\mathcal{P}}_{i,S=1}$, where $i$ runs over all lattice sites, to lowest order in $J/J_H$ the SO(6) "Heisenberg" Hamiltonian becomes:

$$
\hat{H}_{S=1} = \hat{\mathcal{P}}_{S=1}\hat{H}\hat{\mathcal{P}}_{S=1} = J \sum_{\langle ij \rangle} \sum_{m,n=1}^{3} \hat{A}_i^{mn}\hat{A}_j^{mn} = J \sum_{\langle ij \rangle} \mathbf{S}_i \cdot \mathbf{S}_j.
\tag{60}
$$

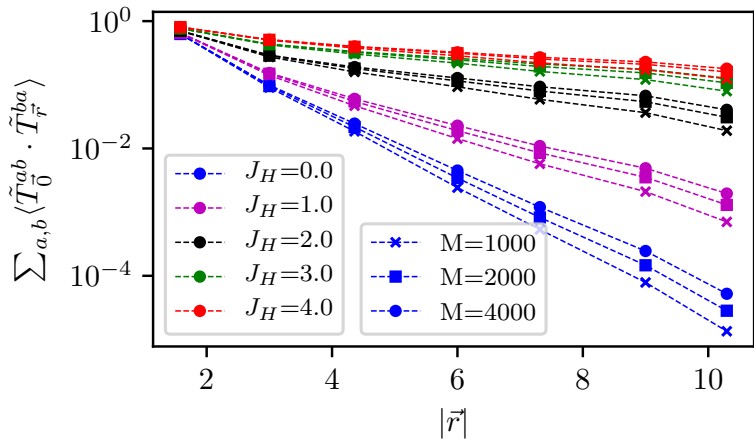

Figure 4: Flavor-flavor correlations obtained using DMRG and shown on a logarithmic scale for a cylinder of circumference $N_y = 3$ and length $N_x = 12$. Different colors correspond to different values of $J_H$ (given in units of $J$) and different markers to different bond dimensions $M$. For $J_H = 0$ the decay of the correlations is consistent with an exponential.

## C  Additional numerical results

### C.1  Probing magnetic order

To complement the analysis presented in Sec. 3.3 of the main text, indicating that a finite Hund's coupling, $J_H$, is required to drive the system into a magnetically ordered state, we calculate flavor-flavor correlations in the absence of pinning fields at the ends of the cylinder. More specifically, the correlations calculated are $\sum_{a,b=1}^{4} \langle \tilde{T}_{\vec{0}}^{ab} \tilde{T}_{\vec{r}}^{ba} \rangle$, where $\vec{0}$ denotes the origin which we choose to be at the left end of the cylinder, and we consider positions $\vec{r}$ on the lattice which correspond to the same sub-lattice as the site at $\vec{0}$ when 120°-order is present. When more than one site on the lattice correspond to the same distance $|\vec{r}|$, a symmetrization is performed and an average value for the correlations is used. Resulting correlations are shown in Fig. 4 for cylinders of circumference $N_y = 3$ as $J_H$ is increased and the bond dimension is varied. For the maximal bond dimension used of $M = 4000$, the truncation error was of order $10^{-4}$.

### C.2  Projection of the SU(4) spin model onto the subspace of singlet coverings using MPS

As mentioned in the main text, to study the projection of the SU(4) Heisenberg model onto the subspace of nearest-neighbor singlet coverings, for system sizes of 12 sites and larger, we use MPS-based simulations.

We start by constructing the MPS representations of the nearest-neighbor singlet coverings. We note that the tensor product of two 6-dimensional vector representations of SO(6) is given by the sum of a symmetric traceless, antisymmetric and a one-dimensional representation (the singlet state). Therefore, the projection onto the singlet state, given by the operator $\hat{P}_{ij}$ (see

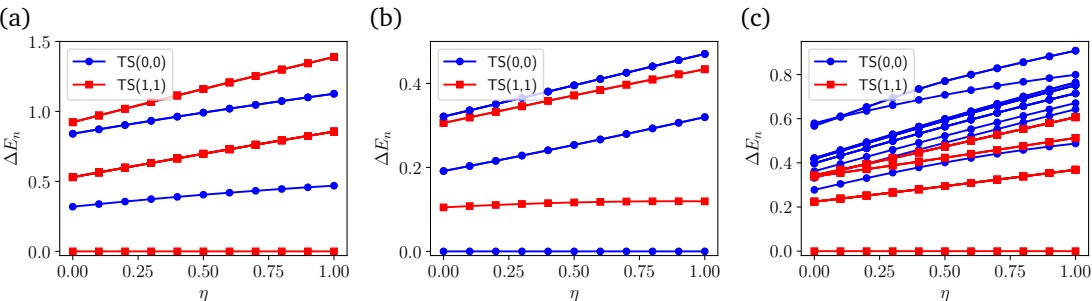

Figure 5: Low energy excitation spectrum of the interpolated Hamiltonian $(1-\eta)H_0 + \eta H_2$ for systems with $N = 12, 16, 36$ sites in (a,b,c) respectively. Energy states in the topological sector TS$(0,0)$ are plotted in blue (circles), and the ones in TS$(1,1)$ are plotted in red (squares).

Eq. (8) in the main text), can be written as

$$
\hat{P}_{ij} \propto \left( \sum_{a,b=1}^{4} \tilde{T}_i^{ab} \tilde{T}_j^{ba} + \hat{\mathrm{Id}}_{ij} \right) \left( \sum_{a,b=1}^{4} \tilde{T}_i^{ab} \tilde{T}_j^{ba} - \hat{\mathrm{Id}}_{ij} \right) =
$$
$$
\left( -\hat{Q}_{ij} + \hat{\Pi}_{ij} + \hat{\mathrm{Id}}_{ij} \right) \left( -\hat{Q}_{ij} + \hat{\Pi}_{ij} - \hat{\mathrm{Id}}_{ij} \right), \tag{61}
$$

where the first (second) term in the product above projects out the anti-symmetric (symmetric) representation. Given a nearest-neighbor covering $C = \{(i_k, j_k)\}_{k=1,..,N/2}$, as was defined in the main text, we can obtain the corresponding singlet covering MPS by applying the matrix product operator (MPO) representation of the product of projectors $\prod_{k=1}^{N/2} \hat{P}_{i_k j_k}$ to a random initial MPS. Note that to allow for an $SU(4)$ singlet covering state on a system of width $N_y$ a bond dimension of $6^{N_y}$ is required for the MPS. Once the MPS representations of the singlet coverings are obtained, both the overlap matrix, required to solve the generalized eigenvalue problem, and the matrix elements of the projected Hamiltonian can be computed. For the latter, an MPO representation of the original spin Hamiltonian is used. We then solve the generalized eigenvalue problem (since the dimension of the projected Hamiltonian is greatly reduced compared to the one of the original spin Hamiltonian, it can be easily diagonalized using standard sparse diagonalization), both to find the ground state of the projected Hamiltonian in terms of the singlet coverings, and to calculate the gap in the projected problem.

To calculate the overlap of the ground state of the projected Hamiltonian with the ground state of the original spin Hamiltonian, we obtain an MPS representation of the latter using DMRG. For the results presented in Table 1 in the main text, bond dimensions used for the calculation of the ground state were between $M = 1000$ and $M = 2000$ depending on system size, resulting in truncation errors $\epsilon$ smaller than $2 \cdot 10^{-3}$ in all cases. A finite truncation error gives rise to an error in the calculation of the overlap that we estimate to be of order $\sqrt{\epsilon}$.

## C.3 Exact diagonalization of the dimer model

### C.3.1 Energies and excitation spectrum of the interpolated Hamiltonian

In Table 3 we summarize the energies of the lowest energy states of the dimer Hamiltonians $H_{n=0,1,2}$ (where $n$ denotes the order of the expansion in $x$) obtained using ED. We list the energies of the lowest energy states in the topological sectors TS$(0,0)$ and TS$(1,1)$, for three different system sizes with $N = 12, 16$ and $36$ sites. We note that the energies obtained for $H_0$ reproduce the ones presented in [21] for $v/t = 0$.

Table 3: Energies of the lowest energy states in the two topological sectors, as well as the gap $\Delta E = E_{(0,0)} - E_{(1,1)}$, obtained using ED of the dimer models $H_n$ (where $n = 0, 1, 2$ is the order of the expansion in the parameter $x = 1/6$) for three different system sizes.

| N=12 | $H_0$ | $H_1$ | $H_2$ |
|---|---|---|---|
| (0,0) | -4.05317 | -3.25070 | -3.29407 |
| (1,1) | -4.37228 | -3.64575 | -3.76333 |
| $\Delta E$ | 0.31911 | 0.39505 | 0.46926 |
| N=16 | $H_0$ | $H_1$ | $H_2$ |
| (0,0) | -5.52971 | -4.43419 | -4.54785 |
| (1,1) | -5.42488 | -4.32630 | -4.42862 |
| $\Delta E$ | -0.10482 | -0.10789 | -0.11923 |
| N=36 | $H_0$ | $H_1$ | $H_2$ |
| (0,0) | -11.76017 | -9.40533 | -9.59950 |
| (1,1) | -12.03778 | -9.91507 | -10.08708 |
| $\Delta E$ | 0.27761 | 0.50974 | 0.48758 |

### C.3.2 Dimer-dimer correlations

In Fig. 6 we present side by side the real space dimer-dimer correlations for the lowest energy state of $H_0$ and $H_2$ respectively, in the two topological sectors TS(0,0) and TS(1,1). As was mentioned in the main text the correlations in TS(0,0) become more uniform for $H_2$, while for TS(1,1) the correlations remain practically unchanged.

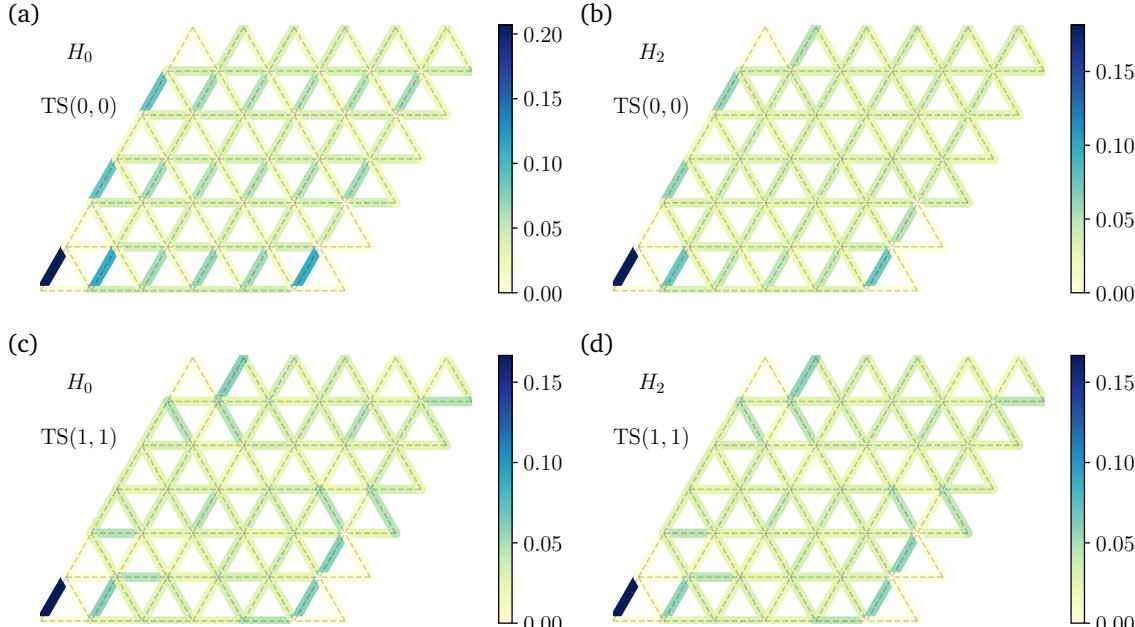

Figure 6: Dimer-dimer correlations $\langle b_1 b_i \rangle$, in the lowest energy state in the topological sectors TS(0,0) and TS(1,1) respectively for the standard dimer model with $v/t = 0$, $H_0$, on the left, and of the extended dimer model $H_2$, on the right.

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
