# Peer review of "Dimer description of the SU(4) antiferromagnet on the triangular lattice"

_SciPost Physics, doi:SciPost Phys. 8, 076 (2020)_

## Round 1 · Referee Report · Anonymous (Referee 1) · 2019-12-23

Strengths

1-Complex study of a SU(4) symmetric Heisenberg model relevant for the spin-orbital system, cold atoms, and for the Mott insulating state of the twisted bilayer graphene using analytical and numerical techniques including DMRG and exact diagonalization of the original Heisenberg model and the effective Hamiltonian in the singlet-dimer basis.

2-After introducing the Hund's coupling, the symmetry of the system breaks down to SU(2) X SU(2). The numerical DMRG study identified a phase transition between the 3-sublattice ordered state of the large Hund's coupling and a spin-disordered state at small values of the Hund's coupling (see Fig. 1) by a finite-size scaling.

3-It is preferable for two 6-dimensional nearest neighbor SU(4) spins to form a singlet dimer, and a good variational ground state appears to be a linear combination of the singlet dimer coverings (a so-called SRVB, short-range resonating valence bond state). The authors have derived the effective model including up to four dimer exchanges in the reduced Hilbert space of the dimer coverings. They have checked that the overlap between the ground state wave functions of the original Heisenberg model and the effective model is significant (Table I).

4-Identification of a translational symmetry breaking spin disordered state as the ground state of the effective model, which can be described by the simple Rokshar-Kivelson like Hamiltonian including the basic off-diagonal term describing the resonance between two dimers.

Weaknesses

1-The precise nature of the dimer ordering in the 12-site unit cell is not revealed. The ordering is not at all apparent in the plot of the dimer-dimer correlations in real space (Fig. 3 a and c). The translation symmetry breaking is identified by the peak in the Fourier transform of the dimer-dimer correlations, however without a finite-size scaling analysis (there is no information about the dimension of the reduced Hilbert space of the dimer coverings, so it is hard to judge the difficulty of diagonalizing larger clusters).

2-The dimer ordering pattern could have been tried by DMRG calculations using pinning fields to set up dimer configurations at the boundary (analogous to the calculation that was used to identify the 3-sublattice spin order). Certainly, this may not work if the ordering is not the ordering of dimers, but is some kind of resonant dimer state, like in the case of the RK model on the square lattice.

Report

The physics of the self-conjugate irreducible representations of the SU(N) symmetric Heisenberg models have been studied extensively for the bipartite square lattice using Quantum Monte Carlo in the earlier literature, as it can present a good example of the resonating valence bond physics of Anderson, and the (singlet) dimers become more stable with increasing N. In this manuscript, the authors extends studies of the self-conjugate 6-dimensional irreducible representations of the SU(4) symmetric Heisenberg models to the frustrated triangular lattice. They are motivated by spin-orbital problems, cold-atom physics, and the fascinating physics of the twisted bilayer graphene, where the valley indices originating from the underlying honeycomb lattice provide the additional flavors.

The authors use analytical and numerical tools to extract the properties of the model. Using DMRG, they show that the flavor-flavor correlations decay exponentially for the pure SU(4) Heisenberg model, however, a three-sublattice ordered state is stabilized by the introduction of the Hund's coupling. For the physics of the pure Heisenberg model, they derive an effective model based on singlet dimers in a systematic manner and compare the applicability of the model by looking at the overlap between the ground state wave functions of the effective model and the original Heisenberg model. Finally, they pursue the nature of the ordering of the dimers in the effective model. They found evidence for a translational symmetry breaking and a root 12 X root 12 order, however, the precise nature of the ordered state remains concealed.

This is a timely study, well presented, and I would like to recommend the publication of the manuscript.

Requested changes

1-In Eq. (2), the usage of the numbers 1...4 for the SU(4) basis states and 1...6 for SO(6) basis is confusing. Would it be possible to use e.g. letters to denote the four flavors of the SU(4) fundamental irreducible representation?

2-The SU(4) versus SO(6) was also discussed in Phys. Rev. B 80, 064413 - perhaps worth comparing and adding a reference.

3-In Sec. 3.3.1. the flavor gap is discussed. Could the authors identify the irreducible representation corresponding to the (t3,t8,t15) = (2,0,0) sector?

4-Perhaps it would be useful to split H_eff in Eq.(37) as H_eff = H_0 + H_1 + H_2 (as it is discussed later on), and to write the H_0, H_1, and H_2 terms separately. It is then easier for the reader to recognize which terms are present for the calculations in Fig. 5.

5-Figure 3: Plase add the system size in the caption, as well as the method (e.g. ED of the effective model).

6-The k values make a square grid in Fig. 3(b) and (d), however, they should make a triangular lattice. So it is not obvious how the k values are defined. Drawing the Brillouin zone would certainly help the reader.

7-Why are there only two peaks in the TS(1,1) sector in Fig. 3(d), is the expected 6-fold rotational symmetry broken by the selection of the topological sector (are the rotated counterparts in the TS(1,0) and TS(0,1) sectors)? If yes, please add this information.

8-Caption of figure 4: I guess the plot shows the *exponential* decay of flavor-flavor correlations. Please modify the caption accordingly. Please indicate the numerical method of how the data were calculated (DMRG ? ).

---

## Round 1 · Referee Report · Anonymous (Referee 2) · 2020-1-9

Report

Report on manuscript 1911.03492 by A. Keselman, L. Savary and L. Balents.

In this manuscript, the authors investigate the SU(4) model on the triangular lattice with the antisymmetric 6-dimensional representation at each site using a classical approximation, DMRG simulations on cylinders, a dimer expansion à la Rokhsar-Kivelson, and exact diagonalizations. While the classical version of the model develops 3-sublattice long-range order, they provide solid numerical evidence with DMRG that the quantum ground state is actually disordered. To further identify the quantum ground state, they derive an effective quantum dimer model up to loops of length 8, for which they find evidence of a VBS with a 12-site unit cell with the help of exact diagonalizations.

The paper is very clear, the problem is interesting and timely, the approaches are all relevant, and the conclusions are well motivated. Nevertheless, I have a number of comments and suggestions, some of them optional.

1) The classical state which, if I got it right, is the best product wave-function, is discussed only in terms of the SO(6) description. I think it would be useful to discuss this wave-function in the SU(4) language, for instance using the fermionic representation of Eq. (2).

2) Having found an ordered classical ground state, it is natural to wonder if quantum fluctuations destroy this order. This is done numerically using DMRG, and the evidence that there is no LRO is strong. Still, it would be nice to have semi-classical results using some kind of flavour-wave theory. I don’t think that this is absolutely needed however, and it might be quite involved, so this is left as an optional suggestion.

3) On page 15, the authors quote the critical value v/t around 0.7 for the occurence of the RVB phase in the QDM on the triangular lattice. This value is consistent with the pioneering work of Moessner and Sondhi, but subsequent investigations (Ref. 21 and two further papers by the same authors, PRB 74, 134301 and PRB 76, 140404) pinned it down to a significantly larger value, 0.83±0.02.

4) To discuss the nature of the quantum ground state, the authors derived an effective Quantum Dimer Model by projecting the model onto the nearest-neighbour singlet dimer subspace. It is easy but cumbersome for a reader to compare the values of the coupling constants for the physically relevant values of alpha, beta and x. The authors might consider including the actual values one way of the other, for instance in a Table.

5) The conclusion that there is VBS order in the ground state of the QDM model is apparently not supported by DMRG results. I wonder if this might be a consequence of the way the dimer model has been derived. As shown in PRB 90, 100406 and PRB 97, 104401, the derivation à la Rokshar-Kivelson, even if pushed to infinite order as in Ref. [19], can still be quite inaccurate because terms generated by virtual off-diagonal couplings outside the short-range dimer subspace can change quite significantly the effective coupling constant of the loop processes inside short-range dimer subspace. For the spin-1/2 kagome model studied in these papers, this changes the nature of the ground state qualitatively. As for point 2), I leave this as an optional suggestion because it is certainly quite involved to go beyond the derivation à la Rokshar-Kivelson. Still the authors might want to comment on this point. Maybe for some reason these off-diagonal processes are less important here than for the spin-1/2 kagome antiferromagnet. But maybe there are important, and the actual QDM is in the RVB phase, which would be quite interesting.

---

## Round 2 · Author Response

We thank the referees for the careful reading of our paper and for their comments and suggestions.
We are providing a revised version of the manuscript addressing all the questions and comments raised.

---

## Round 2 · List of Changes

1. In Eq. (2), and throughout the manuscript, we now use numbers 1..4 in sanf-serif font to denote the SU(4) basis states, as opposed to the standard font used to denote the SO(6) basis states.

2. We have added a reference to Phys. Rev. B 80, 064413 addressing the mapping between the SU(4) and SO(6) groups in the beginning of Sec. 2.2.

3. When discussing the flavor gap in Sec. 3.3.1, we now state that the (t3,t8,t15) = (2,0,0) sector corresponds to the 15-dimensional irreducible representation, as can be verified by calculating the value of the SU(4) quadratic Casimir operator.

4. We have added the system size and the method used to obtain Fig. 3 in its caption.

5. We have modified Fig. 3(b,d) to show the hexagonal Brillouin zone.

6. We have added a discussion regarding the breaking of the 6-fold rotational symmetry down to a 2-fold one, observed in Fig. 3(d) in Sec. 4.4.2. This symmetry breaking occurs because we peak only bonds with a specific orientation when calculating the bond-bond correlations.

7. In the effective dimer model in Eq. (37) we now label all the coefficients and list the corresponding expressions in terms of the parameters (x,\alpha, \beta), as well as their numerical values obtained for the SU(4) Heisenberg model (x=1/6,\alpha=-1,\beta=1) in Table 2. We hope this also make it easier to see which terms appear at which order in the expansion in x. In addition, when referring to H_0, H_1 and H_2 in Sec. 4.4.2, we define them explicitly in terms of these coefficients and their numerical values.

8. We have modified the caption of Fig. 4 to state that it was obtained using DMRG and is consistent with an exponential decay of the correlation function for J_H=0.

9. We have added Appendices (A.2) and (A.3) discussing the classical variational wavefunction in terms of the SU(4) basis states and relating the SU(4) formulation to the SO(6) formulation presented in (A.1).

10. We have fixed the critical value for v/t mentioned in Sec. 4.4 to be 0.83 as pointed out by the referee, and have added references to PRB 74, 134301 and PRB 76, 140404 where this value is obtained.

You are currently on this page

Resubmission 1911.03492v2 on 10 April 2020

---

## Editorial Decision

published